# Mutation Patterns of Human SARS-CoV-2 and Bat RaTG13 Coronavirus Genomes Are Strongly Biased Towards C>U Transitions, Indicating Rapid Evolution in Their Hosts

**DOI:** 10.3390/genes11070761

**Published:** 2020-07-07

**Authors:** Roman Matyášek, Aleš Kovařík

**Affiliations:** Laboratory of Molecular Epigenetics, Institute of Biophysics, Academy of Sciences of the Czech Republic, Královopolská 135, 61265 Brno, Czech Republic; matyasek@ibp.cz

**Keywords:** evolution, cytosine deamination, CpG depletion, coronavirus, SARS-CoV-2, mutation bias

## Abstract

The pandemic caused by the spread of SARS-CoV-2 has led to considerable interest in its evolutionary origin and genome structure. Here, we analyzed mutation patterns in 34 human SARS-CoV-2 isolates and a closely related RaTG13 isolated from *Rhinolophus affinis* (a horseshoe bat). We also evaluated the CpG dinucleotide contents in SARS-CoV-2 and other human and animal coronavirus genomes. Out of 1136 single nucleotide variations (~4% divergence) between human SARS-CoV-2 and bat RaTG13, 682 (60%) can be attributed to C>U and U>C substitutions, far exceeding other types of substitutions. An accumulation of C>U mutations was also observed in SARS-CoV2 variants that arose within the human population. Globally, the C>U substitutions increased the frequency of codons for hydrophobic amino acids in SARS-CoV-2 peptides, while U>C substitutions decreased it. In contrast to most other coronaviruses, both SARS-CoV-2 and RaTG13 exhibited CpG depletion in their genomes. The data suggest that C-to-U conversion mediated by C deamination played a significant role in the evolution of the SARS-CoV-2 coronavirus. We hypothesize that the high frequency C>U transitions reflect virus adaptation processes in their hosts, and that SARS-CoV-2 could have been evolving for a relatively long period in humans following the transfer from animals before spreading worldwide.

## 1. Introduction

The pandemic caused by the spread of SARS-CoV-2 has led to considerable interest in its evolutionary origin and genome structure. Its ~30 kb-long positive-sense single-stranded RNA genome is AU-rich (62%) and encodes 15 proteins [1], preferring pyrimidine-rich codons to purines [2]. The spike protein of SARS-CoV-2 contains a domain important for the contact with the surface angiotensin converting enzyme 2 (ACE2) in human cells [3,4]. Untranslated regions are short, mostly limited to 5’ and 3’ termini, and generally do not exceed 3% of the virus genome. Phylogenetically, SARS-CoV-2 is closely related to *Rhinolophus affinis* (a horseshoe bat) virus, strain RaTG13 (96% identity) [1], and the *Malaysian pangolin* coronavirus (91% identity) [5]. Most epidemiological and sequence data suggest that the primary transfer occurred from bats to humans [6], while the timing and place of the transfer remain a topic of intense debate. The GenBank contained more than 1000 entries of completely sequenced SARS-CoV-2 genomes to date (27 April 2020).

Cytosine appears to be the least stable base in nucleic acids due to deamination to uracil [7]. In single-stranded molecules, the half-life of any specific cytosine is estimated to be about 200 years [8]. Consequently, many genomes, including those of viruses, exhibit relatively low GC content, particularly in areas under low selection constraints, such as various repeats and pseudogenes. The CpG dinucleotides have long been observed to occur with a much lower frequency in the sequence of vertebrate genomes than would be expected due to random chance [9,10]. The depletion is explained by cytosine methylation, which appears to be concentrated to CpG dinucleotides [11] and which deaminates into T. However, some pathogenic viruses (both RNA and DNA), including flu, papilloma, polyoma, and human immunodeficiency virus (HIV), also show a reduction in CpG dinucleotides in their genomes [12,13,14,15], suggesting that the reduction in CpG content may not be limited to nuclear genomes. The C deamination is also believed to contribute to an excess of transition over transversion substitutions in both prokaryotic and eukaryotic genomes [16,17,18,19], although this bias does not seem to be universal and notable exceptions exist [20]. The transition/transversion ratios correlate negatively with evolutionary time, a trend that was more pronounced for rapidly evolving RNA viruses than slowly evolving DNA viruses [21].

In this work, we address the following questions: (i) What are the mutation patterns in SARS-CoV-2 and closely related RaTG13? (ii) Which substitution types contribute to amino acid divergence in the critical ACE2 binding domain? (iii) Does CpG depletion occur in SARS-CoV-2 and other human and animal coronaviruses? We used bioinformatic analyses carried out on publicly available sequences in the GenBank. Evidence was obtained that mutation trends are similar in both SARS-CoV-2 and RaTG13, biased towards C>U and potentially influenced by cytosine deamination processes in their hosts.

## 2. Material and Methods

### 2.1. Source of Sequences

Sequences were retrieved from the GenBank. Strains, GenBank accessions and other details are listed in Appendix A. Only complete sequences with no or few unspecified nucleotides (Ns) were considered for the analysis. Furthermore, human genomes with unrealistically large (>30) numbers of single-nucleotide variations (SNVs) were removed from the datasets.

### 2.2. Identification of Sequence Variants

CLC Genomics Workbench 11.1 (Qiagen GmbH, Hilden, Germany) (CLC) was used to estimate sequence variation between coronaviruses. Briefly, a whole SARS-CoV-2 sequence (Wuhan-Hu-1, GenBank accession number MN908947; previously called NC_045512) was mapped to the RaTG13 (GenBank accession MN996532) accession as the reference. Mapping parameters were set in order to minimize short gaps in the alignments: insertion opening cost_6, insertion extension cost_1 and deletion cost_1. SNVs were named using the following parameters: genome coverage_1; counts_1; frequency_1. Nucleotide composition of the genomes were calculated using the Seqtk_comp tool on a Galaxy server [22] (toolshed.g2.bx.psu.edu/repos/iuc/seqtk/seqtk_comp/1.3.1 developed by Heng Li at the Broad Institute). The accuracy of mapping was confirmed by a pairwise comparison and divergence computations in CLC. In a population-level study of genetic variation, each of the 33 SARS-CoV-2 completely sequenced genomes was mapped to the SARS-CoV-2 reference Wuhan-Hu-1 genome (MN908947). SNVs were named using the same parameters as above. There were 0‒10 SNVs in different SARS-CoV-2 genomes. The data files in the csv format were exported to MsExcel and further processed using program functions (e.g., “countif”, “sum”, “count2”).

### 2.3. Analysis of Data

To express genetic variations between human isolates, data from all genomes were pooled and treated as a single dataset. Indels were not considered. The substitution rate (*R*_i>j_) for each nucleotide was calculated as follows:(1)$$R_{i>j}=\frac{O_{i>j}}{E_{i>j}},$$
where *O*_i>j_ and *E*_i>j_ are the observed and expected frequency of nucleotide changes, respectively, from nucleotide i to nucleotide j. The observed frequency was computed according to the formula:(2)$$O_{i>j}=\frac{S_{i>j}}{S_{\mathrm{tot}}}\times 100_{,}$$
where *S*_i>j_ is the numbers of substitutions of i > j type (e.g., C>U) and *S*_tot_ is the sum of all substitutions. The expected frequency *E*_i>j_ of a nucleotide change from i to j, normalized to the nucleotide content, was computed by:(3)$$E_{i>j}=\frac{N_{i}}{3_{L}}\times 100_{,}$$
where *N*_i_ is the number of i-nucleotides (e.g., C) in the genome and *L* is the genome size in nt. Each of four nucleotides (A, C, G, and U) is likely to change into three different nucleotides reflected by a constant of “*3*” in the denominator. The substitution rates were expressed as a ratio between observed (*O*_i>j_) and expected (*E*_i>j_) frequencies.

The CpG depletion was expressed as the ratio of observed and expected counts of CpG dinucleotides, calculated according to the formula:(4)$$CpG_{\mathrm{depletion}}=\frac{N_{\mathrm{CpG}}\times L}{N_{C}\times N_{G}},$$
where *N*_CpG_, *N*_C_, and *N*_G_ are the numbers of CpG dinucleotides, *C*, and *G* nucleotides, respectively, and *L* is the sequence length.

Synonymous and nonsynonymous mutations in SARS-CoV-2 reading frames were determined according to annotated accession MN908947. Nucleotide diversity values were calculated using the DNASp4 software (University of Barcelona, Spain) [23]. Amino acids’ hydrophobicity values were taken from the https://www.cgl.ucsf.edu/chimera/docs/ server [24] corresponding to those experimentally determined by [25]. Multiple alignment of SARS-CoV-2 genomes was carried out using ClustalW program built-in the SeaView4 software [26]. Aligned sequences were trimmed from both ends by about 40 nucleotides to get the best match and to remove sequencing artifacts. The phylogeny tree was calculated using a maximum likelihood algorithm implemented within SeaView4. Gaps were ignored.

An online tool was used to construct box plot graphs (http://shiny.chemgrid.org/boxplotr/) [27]. Boxes represent the second quartile, median and third quartile; the whiskers show maximum and minimum values. Correlation analyses were performed using SPSS (Statistical Package for the Social Sciences) 16.0 software (SPSS Inc., Chicago, IL, USA).

## 3. Results

### 3.1. Sequence Comparison and SNP Analyses of Related SARS-CoV-2 and RaTG13 Genomes

We compared the frequency of mutations in SARS-CoV-2 (MN908947), using the bat RATG13 (MN996532.1) as a reference. The SARS-CoV-2 and RaTG13 were 29903 and 29855 nt long, respectively. Compared to RATG13, there was one deletion and seven insertions in the SARS-CoV-2 genome; the longest insertions spanned a 12 nt-long GC-rich region between nucleotides 23601 and 23612. Nucleotide counts differ between the RaTG13 and SARS-CoV-2 genomes, with the latter containing fewer Cs, whereas the contents of other nucleotides were enriched (Figure 1a and Appendix A). The pyrimidine and purine residue levels were balanced in both genomes. There were 1136 SNPs between RaTG13 and SARS-CoV-2, resulting in about a 4% divergence, consistent with the previous study [1]. The characteristics of single-nucleotide variations (SNVs) are shown in Figure 1b; a summary of mutation analysis is given in Appendix A. It is evident that the C>U and reverse U>C transitions were by far the most frequent mutations, accounting for 60% of all SNVs. The G>A and reverse A>G transitions were the next most abundant, though their frequency was substantially lower (23%). The rate of C>U substitutions was higher (5.01) than that of reverse U>C (2.73) substitutions (Appendix A). These transitions were more abundant than those of A>G and G>A (Figure 1e). The G>C and U>G substitutions had the lowest rates, with observed to expected ratios of 0.07 and 0.13, respectively. When considering all substitutions, the overall frequency of transitions outnumbered the frequency of transversions by almost 10-fold (Figure 1d). Two other randomly selected isolates of human coronavirus from 2019‒2020 yielded essentially the same results, since there is no or little variation among the genomes.

### 3.2. Characteristics of SARS-CoV-2 Virus Variants

We analyzed 34 fully sequenced SARS-CoV-2 coronaviruses isolated from different populations (Appendix A), representing worldwide virus diversity. Phylogenetic relationships between the sequences are revealed by a maximum-likelihood tree (Figure 2) forming three relatively well supported branches. The L and L’ branches contained sequences previously characterized as the “L” genotype [28]; the S branch contained the less abundant “S” genotype. Each of the genomes was mapped to the Wuhan-Hu-1 reference genome and SNVs were counted in pairwise alignments.

The number of SNVs per genome ranged from zero (MT253696 from China) to 10 (MT263469 from the USA). The datasets contained 152 SNVs, of which 84 (55%) were shared between at least two accessions; 68 (45%) were singletons (Appendix A, sheet 1). After the subtraction of duplicities (i.e., counting each SNV just once), the remaining number of SNVs was 85. We determined the mutation characteristics in these variants (Figure 1c and Appendix A, sheet 2). Similar to the previous bat to human comparison, the C>U and U>C transitions predominated in the SARS-CoV-2 interpopulation comparisons. Mapping of SNV types to phylogeny showed a high proportion of variants bearing C>U substitutions in both S’- and L-type groups (Figure 2 and Appendix A, sheets 5-7). The highest (5.19) observed/expected ratio was found for the C>U substitutions, while the reverse U>C type had a markedly lower ratio (1.32). No A>C substitutions were found and the ratios for A>U were low (0.11). The transitions/transversion ratio was somewhat lower than in the case of bat versus SARS-CoV-2 comparisons (Figure 1d).

Each SNV was further analyzed to see whether it does (nonsynonymous mutation) or does not (synonymous mutation) change the amino acid sequence (Appendix A). Out of the 27 C>U substitutions, 14 (52%) were nonsynonymous, 11 (41%) were synonymous, and two (7%) occurred in untranslated regions; out of the 12 reverse U>C substitutions, four (33%) were nonsynonymous and eight (67%) were synonymous. We also determined the extent to which changes affected the hydrophobicity of the replaced amino acid residues (Figure 3 and Appendix A). Compared to other substitutions, the replacement of C with U nucleotides significantly elevated the frequency of hydrophobic codons (one-way ANOVA, *p* < 0.001), while U>C substitutions reduced it (*p* < 0.05).

### 3.3. Variation in the Surface Glycoprotein (Spike) Subregion

The spike protein harbors a domain binding the ACE2 receptor on the cell surface, believed to be important for host‒virus interactions. Comparison of RaTG13 and SARS-CoV-2 showed a relatively even distribution of SNVs, except for a prominent peak located within the surface glycoprotein subregion (Figure 4a), where a 153 nt receptor-binding domain (RBD) [3,6] was the most variable both at the nucleotide (68% similarity) and protein (76% similarity) levels. The alignment of RBD domains between bat RaTG13 and human SARS-CoV-2 is shown in Figure 4b. In this domain, the number of synonymous (silent) and nonsynonymous substitutions was nearly the same, while synonymous substitutions dominated in the rest of the protein (Table 1). The proportion of C>U and U>C transitions relative to other substitutions was 2:3, reflecting the genome average (Figure 4c). Their frequency in nonsynonymous sites was relatively low. Previously, six amino acid residues have been identified [4] that appear to be critical for the virus interaction with the ACE2 receptor (boxes in Figure 4b). Out of these, five residues (Leu486Phe, Tyr493Gln, Arg494Ser, Asp501Asn, and His505Tyr) differed between RaTG13 and SARS-CoV-2. In the altered codons, the frequency of nonsynonymous C>U and U>C substitutions was relatively high (Figure 4d).

### 3.4. CpG Depletion Analysis in Coronaviruses

CpG depletion is a characteristic feature of eukaryotic genomes and some viruses. We determined the frequency of CpG dinucleotide in 17 human (including eight randomly selected SARS-CoV-2 accessions), nine bat, and eight other animal *Betacoronaviruses* (Figure 5a). The number of CpGs ranged from 652 to 1766 among the genomes. The lowest number (652) was found in the human NL63 strain (MK334043.1); the highest was in the Japanese bat *Pipistrellus abramus* (1766). The *Rhinolophus affinis* (bat) RATG13 coronavirus had 882 CpGs in its genome, which is four sites more than was found in SARS-CoV-2 (MN908947). Out of 16 dinucleotide types, the frequency of CpG dinucleotide in SARS-CoV-2 had the lowest value (0.015), less than half that of GpC (0.037). Both the SARS-CoV-2 and NL63 strains exhibited the lowest CpG_obs/exp_ levels. The statistical evaluation of data is presented by way of box plots (Figure 5b). None of the comparisons showed significant differences between the groups (one-way ANOVA, *p* > 0.05).

## 4. Discussion

It is well established that mutation biases for C>U, U>C, A>G, and G>A transitions are the most frequent SNVs in eukaryotic and prokaryotic genomes [16,17,19]. The bias towards these types of mutations is, however, exceptionally high between the SARS-CoV-2 and related bat RATG13 genomes since more than 80% of the mutation spectrum can be attributed to these four substitutions despite them representing just one-third of all possible substitutions. The remaining eight types of substitutions (all transversions) represented about 17% of all SNVs. The skewed mutation rates towards transitions are consistent with highly conserved pyrimidine to purine ratios despite the genome divergence. Together, the RNA genomes of bat RATG13 and SARS-CoV-2 exhibit a strong transition‒transversion bias typical for recently diverging virus strains [21].

When comparing bat RaTG13 and SARS-CoV-2, the C>U transitions outnumber those of reverse transitions almost two-fold. This unexpected nonequilibrium may point to a higher probability of C mutation into U in SARS-CoV-2 than U into C. This bias may potentially explain the relatively lower C nucleotide content of SARS-CoV-2 compared to the bat RaTG13 coronavirus. The reference genome (MN908947) to which the rest of the SARS-CoV-2 genomes were compared is assumed to originate from one of the first cases detected in Wuhan province in China, and may be closest to the SARS-CoV-2 ancestor. Most SARS-CoV-2 isolates from different geographical localities exhibited C>U substitutions when compared to the Wuhan-Hu-1. Moreover, phylogenetically divergent S and L’ lineages seem to harbor a higher proportion of C>U and U>C substitutions than the L lineage which contained the Wuhan-Hu-1 reference genome. The observations suggest that a trend towards the loss of Cs may be ongoing during the pandemic.

### 4.1. Contribution of C-Deamination Events to SARS-CoV-2 Mutability

A question arises about the mechanisms underlying the observed SARS-CoV-2 mutation profiles. RNA viruses exhibit very high mutability, living at the edge of extinction [31]. A virus’s mutagenic capability depends upon several factors, including the fidelity of viral enzymes that replicate nucleic acids, the cellular enzymes of the host, and environmental factors. In this context, variants of RNA-dependent RNA polymerase (RdRp) with potentially altered activities have been reported among the SARS-CoV-2 isolates [32]. However, contrast to most other viruses, betacoronaviruses are known to exhibit proofreading capacity [33], suggesting that polymerase errors are unlikely to generate the strong mutation biases we observe here. A more plausible explanation of the skewed mutation patterns (towards C>U) is spontaneous or enzymatic cytidine deamination processes converting Cs into Us. In double-stranded DNA, the deamination of methylated C into T results in the replacement of G with A on the other strand after replication. Thus, in DNA genomes, the loss of Cs is accompanied by a loss of Gs on the opposite strand, maintaining an equal number of both nucleotides. However, coronaviruses spend most of their life in a single-stranded positive RNA form and a complementary negative strand is produced only during virus replication. It is estimated that these negative-strand intermediates are only about 1% as abundant as their counterparts [34]. Consequently, a negative strand of the virus genome is less likely to accumulate the C>U transitions than the positive strand. Hence, G>A substitutions are relatively rare compared to C>U ones (Figure 1), although both transitions are more common than any of the transversions.

The C-deamination rate in a single-stranded nucleic acid equals one cytosine deamination per 200 years [8], which is double that of A and G bases. Considering there are about 5500 Cs in the genome (Appendix A), there might be 28 spontaneous deamination events in the viral genome per year. This would increase the C>U transition rate to about 4 × 10^−3^ per year, which is almost ten-fold higher than the overall mutation rate in the SARS-CoV-2 genome, estimated to be 6 × 10^−4^ nucleotide substitutions per site per year [35]. Perhaps many deamination events actually generate deleterious variants, which are eventually lost by natural selection. In this context, the C>U transitions may increase stop codon frequency over time, reducing virus fitness. This is because the three eukaryotic stop codons (UAA, UAG, and UGA) are 78% AT-rich. It will be interesting to analyze high-throughput reads to determine if these variants occur in a mutant cloud of descendants. Certainly, the process might be accelerated by cytoplasmatic cytosine deaminase activity [36] acting on single- or double-stranded intermediates. Of note, the Apo3G deaminase enzyme apparently inhibited viral activity in human cells [37].

### 4.2. Relationship between SARS-CoV-2 Mutability and CpG Depletion

There might be a relationship between mutation bias, GC content, and the frequency of individual motifs in the genomes. Both the SARS-CoV-2 and bat RATG13 genomes showed a reduced content of CpG dinucleotides, in contrast to most other coronaviruses, which exhibit an enrichment of CpGs (e.g., Middle East Respiratory Syndrome Coronavirus (MERS)) in their genomes. It should be emphasized that the magnitude of CpG depletion in both genomes was much lower than that reported for other human viruses [13]. For example, in some nonintegrating small dsDNA viruses (papillomaviruses and polyomaviruses), the CpG_OBS/EXP_ values were as low as 0.2 [38]. In viruses that integrate into the genome, such as HIV, the reduction of CpGs is explained by 5-methylcytosine demethylation of CpG motifs in integrated copies [12,15]. Thus, it seems that processes operating on RNA might be responsible for general C suppression, whereas processes operating on DNA might be responsible for CpG suppression. The reduction of CpG in SARS-CoV-2 is better explained by structural features of the CpG dinucleotide and a higher tendency to deaminate residing C than in other contexts. It has been proposed that host specificity may play a role in shaping the CpG content [14]. However, in our study, such a relationship is not obvious since we observed a large variation in CpG depletion between human and animal coronaviruses (Figure 5). It is more likely that phylogenetic distance rather than host specificity plays the decisive role. Currently, we can only speculate as to why the human SARS-CoV-2 and NL63 strains exhibit a relatively high level of CpG depletion out of all the coronaviruses analyzed. Interestingly, both strains cause severe pathological effects in their hosts. The CpG-rich DNA has been shown to be immunogenic in humans [39]. Perhaps the pathogenic viruses are under selection constraints to avoid immune systems. Another possibility is that the CpG depletion reflects increased genome mutability, perhaps associated with rapidly spreading new variants. In this context, the HIV retrovirus shows a stronger CpG depletion than the HTLV1 retrovirus, causing leukemia in endemic areas [12]. Only HIV has spread worldwide.

### 4.3. Do C>U Transitions Have Adaptive Value?

In nuclear DNA, abundant C>U and G>A transitions accumulate in sequences under low selection constraints such as the pseudogenes. A biochemical difference in replaced amino acids tends to be greater for transversions than for transitions [40]. Considering the mutations underlying the divergence between bat RaTG13 and SARS-CoV-2, about 70% of C>U transitions occurred in synonymous sites, suggesting their inconsequential effect (neutral or nearly neutral). However, in critical amino acids needed for receptor binding, C>U transitions accounted for almost half of the nonsynonymous mutations. Furthermore, in SARS-CoV-2 variants, some transitions markedly altered the biochemical properties of amino acids. For example, five out of 27 C>U substitutions (19%) involved proline (Appendix A), which is known to be a strong helix breaker. Furthermore, the C>U transitions were strongly skewed towards leucine, isoleucine, and phenylalanine codons, consistent with shifts towards codons to more hydrophobic amino acids after the deamination of cytosine and 5-methylcytosine residues in nucleic acids [41]. Switches between hydrophilic and hydrophobic amino acid residues may have a dramatic effect on protein properties. Thus, the notion of the purely neutral character of C>U mutations might be oversimplified, especially considering their high frequency and global character. As with other mutations, the newly generated variants can respond to selection in variable environments.

Several hypotheses have been proposed to explain the origin of SARS-CoV-2 virus [6,28]: (i) natural selection in the animal host before zoonotic transfer; (ii) natural selection in humans following zoonotic transfer; (iii) the virus is a product of artificial manipulation. Our data make the third possibility less likely since the viral genome seems to follow a similar evolutionary trajectory to that of the closely related bat virus RaTG13. We favor the second hypothesis postulating that cytosine deamination processes are driving the SARS-CoV-2 genome evolution, particularly over short evolutionary times. Resulting C>U biases in mutation spectra could be a stigma of virus adaptation in its host(s).

## Figures and Tables

**Figure 1 genes-11-00761-f001:**
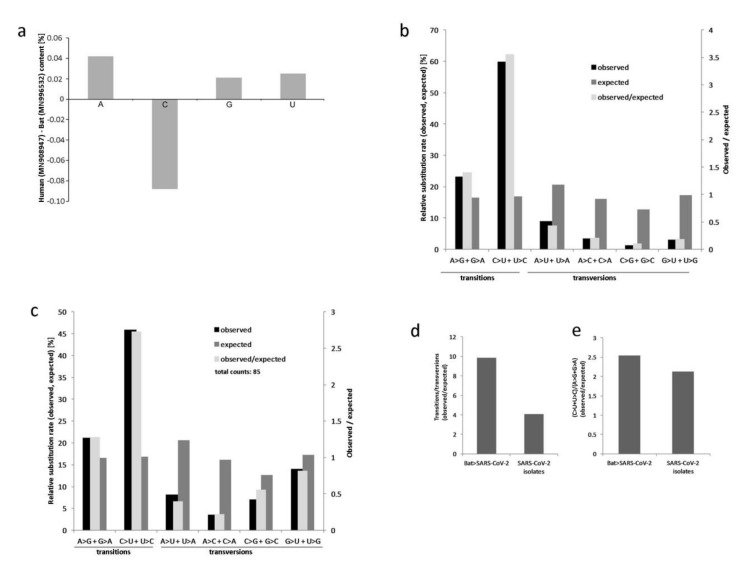
Nucleotide substitution spectra in coronavirus genomes. (**a**) Cumulative frequency of nucleotides in human SARS-CoV-2 genomes. Positive and negative values indicate enrichment and depletion of bases in the SARS-CoV-2 (MN908947) relative to the RaTG13 (MN996532) reference, respectively. (**b**) Nucleotide substitution rates for each of the four nucleotides between RaTG13 and SARS-CoV-2 genomes expressed as observed/expected ratios. The direction of changes is from RaTG13 to SARS-CoV-2. Calculations are according to Equations (1)–(3) in the Methods section. (**c**) SNV characteristics in the human SARS-CoV-2 isolates. The data were pooled from 85 variants identified in 33 genomes (each genome was compared to the reference (MN908947) (Appendix A). Enrichments of transition/transversion (**d**) and (C>U + U>C)/(G>A + A>G) ratios (**e**) between bat RaTG13 and SARS-CoV-2 (MN908947) and among the SARS-CoV-2 isolates relative to corresponding expected values.

**Figure 2 genes-11-00761-f002:**
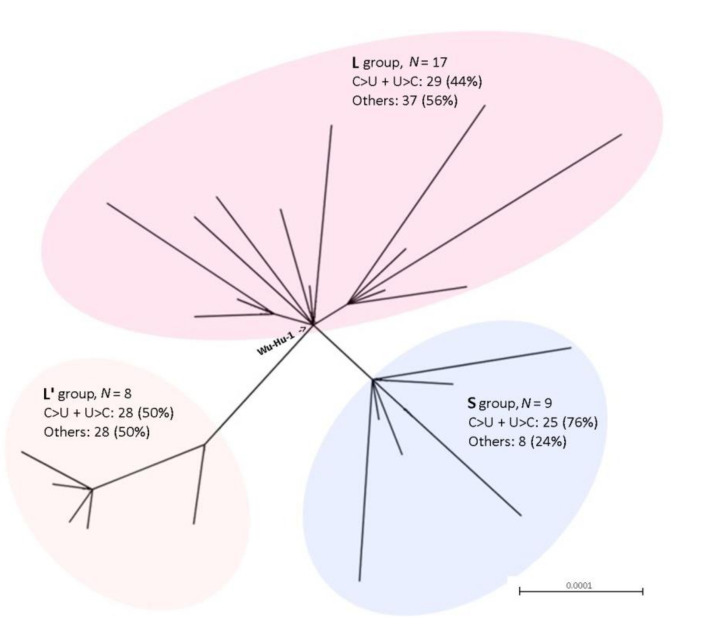
Phylogenetic relationships between human SARS-CoV-2 viruses. The maximum-likelihood tree was constructed from the alignment of 34 whole genome sequences (Appendix A). Statistical support calculated as the approximate likelihood-ratio test [29] was 98% for three branches indicated in colors. The reference Wu-Hu-1 (M908947) sequence fell within the “L” group. *N*—number of sequences in each branch followed by the frequency of substitution type.

**Figure 3 genes-11-00761-f003:**
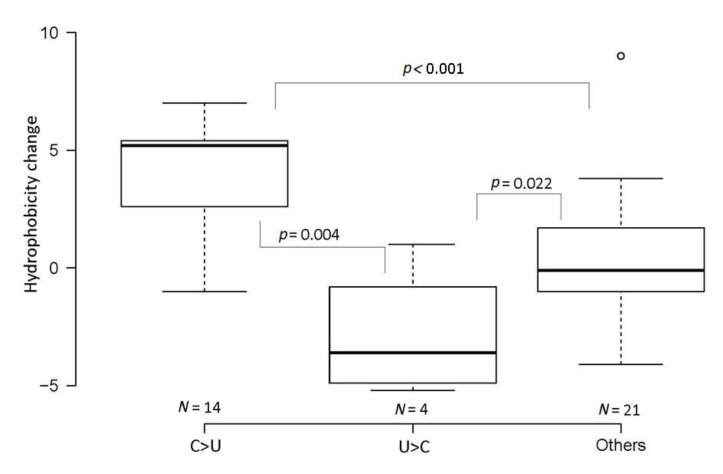
Changes in the hydrophobic properties of amino acids in nonsynonymous substitutions. The Y axis represents shifts in hydrophobicity values between replaced amino acids. The data were obtained from single-nucleotide variations (SNVs) in the SARS-CoV-2 sequences (Appendix A).

**Figure 4 genes-11-00761-f004:**
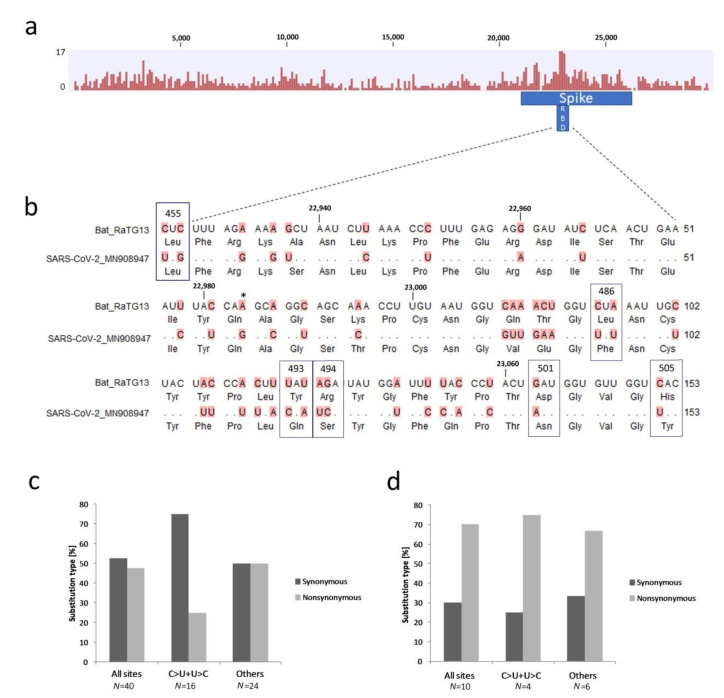
Sequence variation between the bat RaTG13 and Sars-CoV-2 (MN908947) genomes. (**a**) A graphical representation of SNVs. The height of the columns reflects the number of SNVs in a 100-nt sliding window. (**b**) Alignment of the ACE2 receptor-binding domains (RBD). Coordinates are according to a spike protein reading frame in the SARS-CoV-2 genome. The direction of mutations is from the bat RaTG13 to human SARS-CoV-2 sequence. The differences between SARS-CoV-2 and bat RaTG13 sequences are highlighted. Amino acids in a contact zone are boxed. The site at position 22984 (asterisk) was also polymorphic among human SARS-CoV-2 isolates (Appendix A sheet 3). The occurrences of distinct substitutions in synonymous and nonsynonymous sites in (**c**) the whole ACE2 receptor-binding domain (RBD) and (**d**) a contact zone sub-domain.

**Figure 5 genes-11-00761-f005:**
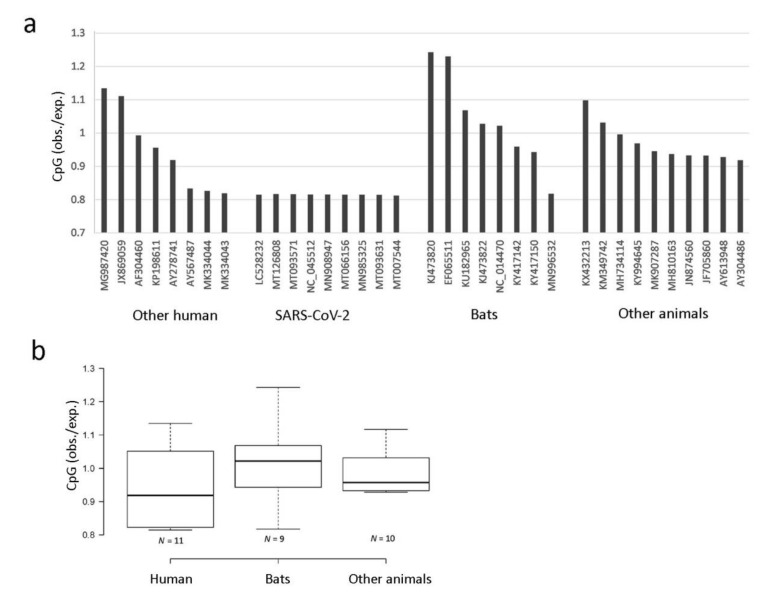
CpG depletion analysis in coronavirus genomes. (**a**) Bar charts showing CpG_OBS/EXP_ in individual genomes. The calculations are according to Equation (4) in the Methods section. (**b**) A statistical representation of CpG depletions in individual groups. The Wuhan-Hu-1 accession (MN908947) was included in the “Human” group as a representative of SARS-CoV-2. Note that, except for RaTG13 (MN996532), bat coronaviruses show no or little CpG depletion.

**Table 1 genes-11-00761-t001:** Analysis of nucleotide diversity of the RaTG13 and SARS-CoV-2 coronavirus envelope s-glycoproteins.

Region ^(1)^	Size (nt)		Synonymous			Nonsynonymous			K_a_/K_s_
	RaTG13	CoV-2	Dif. ^(2)^	Pos. ^(3)^	K_s_ ^(4)^	Dif. ^(1)^	PPos. ^(2)^	K_a_ ^(5)^	
whole	3810	3822	221	888.7	0.3021	40	2915.2	0.0138	0.0457
RBD	153	153	21	36.6	1.0830	19	116.4	0.1841	0.1690
rest	3657	3669	200	855.2	0.2803	21	2798.8	0.0075	0.0268

^(1)^ whole—whole protein, RBD—ACE2 receptor-binding domain; rest—whole protein minus RBD. ^(2)^ Dif., the total number of synonymous or nonsynonymous differences. ^(3)^ Pos., the total number of synonymous or nonsynonymous sites. ^(4)^ K_s_, the number of synonymous (or silent) substitutions per synonymous (or silent) site. Calculated according to Nei and Gojobori [30]. ^(5)^ K_a_, the number of nonsynonymous substitutions per nonsynonymous site. Calculated according to Nei and Gojobori [30].

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
