# Peer review of "Mutation Patterns of Human SARS-CoV-2 and Bat RaTG13 Coronavirus Genomes Are Strongly Biased Towards C>U Transitions, Indicating Rapid Evolution in Their Hosts"

_genes, 2020, doi:10.3390/genes11070761_

Round 1
Reviewer 1 Report
This is a timely and potentially valuable analysis of mutation patterns in the SARS-CoV-2 genome. However, there are several important issues that require major revisions. The methods should be described in more detail; it is often not clear what exactly the data represent. I have some concerns about the statistics used in the paper and the methods used to count mutations. Some of the conclusions do not appear to be supported by evidence. Both the data analysis and the discussion lack depth even when providing additional evidence relevant to the questions addressed in the paper would not be too difficult or time-consuming. There is very little discussion of the present results in the context of existing literature. In addition, there is a large number of typos, omissions, and inconsistencies, some of which I tried to point out below. My overall impression is that the authors rushed this manuscript to submission, which is understandable with respect to current interest in COVID-19. However, I believe at least some additional data analyses and discussion need to be included to support the conclusions. See the specific comments below for details.
Specific comments:
- Abstract: In “can be attributed to C>U and U>T substitutions,” I suspect the U>T is a typo. I believe that standard sequencing methods would not detect U>T substitutions. It might also be helpful to clarify whether these are substitutions in the human strain using the bat strain as a reference or whether it is the other way around (see also other comments below). The following sentence “A similar trend was observed among the sequenced SARS-CoV-2 genomes,” should be removed. It was already stated that it applies to SARS-CoV-2 in general rather than a specific isolate.
- Line 33: “AT-rich” does not seem appropriate for an RNA virus.
- “preferring pyrimidine rich codons to purines” – is this specifically a property of the coding sequences or also the noncoding segments? This is important with respect to possible mechanisms leading to this preference.
- Line 75: A proper reference should be included. See the link “Citing Chimera” on the Chimera website.
- Consolidate contradicting statements on lines 62 and 81. One states that SARS-CoV-2 was used as a reference, the other that RATG13 was used as a reference.
- Line 90: How was the chi square test used? What is the input for the test? I am asking because some of the key assumptions for the chi square test are that it is applied to categorical data (raw counts of observations, not percentages) and that all observations are independent. If counts of substitutions were pooled over multiple pairwise comparisons among different genomes then the assumption of independence is almost certainly violated because a single mutation can be observed in comparing different pairs of genomes and those do not represent independent events.
- Line 91: “the overall frequency of transitions outnumbered the frequency of transversions by 5 fold” – how does this ratio of transitions and transversions compare to other types of organisms? Is it unusual?
- Figure 1: It is not clear how the direction of the substitutions was determined. In addition to conflicting statements about what the reference genome is, it is unclear if this is a comparison between RATG13 and a specific isolate of SARS-CoV-2 or some combination of comparisons between RATG13 and all SARS-CoV-2 isolates. I cannot imagine how it was done for panel b, which apparently summarizes many pairwise comparisons. It should probably be symmetrized by combining C>U and U>C substitutions etc. In any case, significant clarifications need to be added to the legend and to the description of the methods.
- Figure 2: “The direction of mutations is from the human to bat sequence.” This is not very important but I think it would make more sense to use the bat isolate as a reference and the direction of mutations from bat to human. It is generally assumed that the virus jumped from bats to humans rather than the other way around, and one might argue that, in the absence of other evidence, the bat strain is likely to be more similar to their common ancestor than the human strains.
- Table 1: Explain what is Ref./Position/Allele and Ref., Allele, and Shift. If Shift is the difference between Ref. and Allele than at least one number is incorrect (the line starting with 565). The legend states that the hydrophobicities come from the Kyte and Doolittle paper but in Methods they are claimed to come from the Chimera server.
- Line 123 and line 92: Why is the ratio of transitions to transversions 5 in one case and 2 in the other? I understand that these are different sets of data with the latter excluding the duplicate mutation counts but the difference can point to difficulties in estimating the mutation ratios accurately. I would expect that better methods of counting the mutations might produce more consistent results. I think these differences require more thought and reevaluating whether the methods are appropriate.
- Line 129: “We also determined the extent to which replacement would affect biochemical properties of replaced amino acids” – this is somewhat misleading because the only property that was investigated was the amino acid hydrophobicity.
- Line 131: “The average change in hydrophobicity value [15] per C>U mutation was plus 3.68 while it was minus 2.85 for the reverse U>C mutation” – how do these values compare to other types of mutations? I am also puzzled why the values are different for C>U and U>C mutations. This, too, might warrant more discussion. The authors might also want to have a look at Tramontano and Macchiato 1986, Nucleic Acids Res.
- The chi square statistics in Figure 3 may be problematic for the same reason I mentioned above. The Sars-CoV-2 genomes are nearly identical and can therefore hardly be considered independent for the purposes of the statistical test. I do not know how closely related the other viruses are but the same issue probably applies at least to some extent to the other groups as well. Applying the test to data that do not represent independent observations can dramatically inflate the perceived statistical significance. Along the same lines, the comparison in Figure 3a might be misleading if some of the groups are more diverse than others. For example, the SARS-CoV-2 genomes all appear consistently at the low side compared to the other groups but that could be a simple consequence of these viruses being much more closely related than those in the other groups.
- Line 158: I would question the phrase “strong transition-transversion bias typical for eukaryotic DNA genomes.” The bias is present in all domains of life and not typical of eukaryotes. Moreover, some viruses have been reported to have particularly strong transition-transversion bias, which might be worth mentioning in the discussion.
- Line 159: The statement “Putative adaptive character of C>U mutations is demonstrated in the ACE2 receptor domain in which three out of five variable amino acids contained C>U signatures in their codons” is not supported by evidence. The authors mentioned that 3 of the 5 codons contain C>U or U>C mutations but left out that 4 of the 5 contain other mutations. Because, in the whole genome, the C>U/U>C accounts for 60% of all mutations, I think it is more reasonable to argue that the variability of the ACE2 receptor domain is mostly driven by other mutations than C>U/U>C. The authors also did not present any evidence that the receptor domain is under positive selection and thus directly involved in the adaptation to human host, although it is reasonable to expect that this is the case. It might be useful to compare Ka/Ks ratios in that region to that in the rest of the gene or in the other genes to support that claim. Performing Ka/Ks ratio analysis separately for C>U/U>C substitutions and other substitutions may also be relevant to how much the C.U/U>C mutations contribute to evolutionary adaptations of SARS-CoV-2 compared to other mutations, or whether they tend to be mostly neutral.
- Line 161: “It is intriguing that a similar mutation trend was observed among 33 human isolates.” I do not see this as particularly surprising. I believe similar trends have been observed in HIV and other human viruses. The authors should include a review of relevant literature and compare this result with previously published data for other viruses.
- Line 166: “the frequency of C>U substitutions was about 2 fold higher than that of reverse U>C substitutions” – I am very skeptical about this statement considering that the authors did not explain how they determined the direction of the mutations and that they apparently count a single mutation multipole times. Notably, the C>U and U>C mutations are about equally frequent in Figure 1a, which again is left out from this argument.
- The argument on lines 170-176 may be problematic. First of all, no evidence is presented to support the statement “the frequency of C>U (U>C) transitions between the SARS-CoV-2 genomes is consistent with spontaneous C deamination events.” Moreover, the estimate of one C deamination per 200 years is not specific to SARS-CoV-2 and the mutation rates could be very different in viruses and cellular organisms, and also very different among different viruses.
- Line 180: “Some transitions markedly altered biochemical properties of amino acids evidencing for a positive selection at specific sites” – that some transitions changed hydrophobicity of the encoded amino acid can in no way be considered evidence of positive selection. Additional evidence is needed to make that claim. A detailed Ka/Ks ratio analysis with appropriate assessments of statistical significance might be helpful in that regard. See Choi and Lahn 2003, Genome Res., for an example how it can be used.
- Line 183: “consistent with shifts towards codons for more hydrophobic amino” – this is another problematic statement. There is no indication that the authors made any attempt to determine the ancestral sequence and the actual direction of the mutation or any particular C-U mismatch in the alignment. Consequently, it is impossible to tell whether the mutation was C>U or U>C and whether the amino acid change was from less hydrophobic to more hydrophobic or the other way around.
- Line 186: “the virus protein products would be modified by ongoing and relatively fast C deamination processes affecting its genome” – as stated above, it is unclear if the C>U mutations contribute to nonsynonymous changes in proteins more than other mutations. At the minimum, the authors should include other types of mutations in Table 1 and compare the synonymous-to-non-synonymous ratios for different types of mutations. There is also still the issue related to uncertainty as to the actual direction of the mutation, which could only be determined if the ancestral allele is known.
- One important thought related to CpG suppression is that, in RNA, the likelihood of C>U substitution should be independent of whether the C is preceded by G or not. The reason why C is much more likely to change to T in some organisms’ DNA if it is preceded by G is that the C is first methylated and a subsequent deamination changes the methyl-C to T. An unmethylated C changes to U upon deamination, and the U is generally enzymatically converted back to C when it is in DNA. Consequently, processes operating on RNA might be responsible for the general C suppression whereas process operating on DNA might be responsible for CpG suppression. In that regard, it might be interesting to include more discussion on how these results relate to the physiology of coronaviruses (or other RNA viruses), particularly with respect to whether they persist in the host in the DNA form.
- More attention might also be given to the observation of elevated A>G and G>A mutations. It suggests a possible mechanism involving C deaminations in the complementary strand, which again can take place only in a double-stranded nucleic acid.
- The final conclusion (lines 216-218) could be more accurately rephrased in the sense that C>U mutations driven by spontaneous cytosine deamination may contribute to rapid evolution of the virus. I believe this is supported by the data in the paper but some of the more specific statements made earlier are not.
- Supplementary tables need explanatory legends.
- I would urge authors to provide more context for each mutation listed in Table S3: whether it is in a gene or intergenic, synonymous or non-synonymous, the amino acid change, …
Typos and other recommended minor corrections:
- Abstract: pandemy -> pandemic; “interest about its evolutionary origin” -> “interest in its evolutionary origin”; italicize “Rhinolophus affinis”; “Contrast to most other coronaviruses both SARS-CoV-2 and RaTG13 exhibited CpG depletion in their genomes” -> “In contrast to most other coronaviruses, both SARS-CoV-2 and RaTG13 exhibited CpG depletion in their genomes”; “The data support that the hypothesis” -> “The data support the hypothesis”
- Line 39: “completely sequence” -> “completely sequenced”
- Line 60: “intragenomic variation between 60 coronaviruses”: if anything, it should probably be “intergenomic” but I would recommend “sequence variation”
- Line 65: “were mapped the reference genome” -> “were mapped [on|to|onto] the reference genome”
- Line 70: “the number of observed CpG/number of expected CpG dinucleotides” -> “the ratio of observed and expected counts of CpG dinucleotides”
- Line 76: “On line” -> “Online”
- Line 84: “resulting to” -> “[leading to|resulting in|reflecting]”
- Line 92: “by 5 fold” -> “5 fold”
- Line 140: “between the genomes” -> “among the genomes”
- Line 156: “despite these represent” -> “despite these representing”
Author Response
Dear Sir/Madame,
We wish to thank you for your highly professional analysis of our work and for your advice with the revision. It is not so common to receive such a detailed and insightful comments these days.
Below, please find step-by-step responses to your comments.
Specific comments:
- Abstract: In “can be attributed to C>U and U>T substitutions,” I suspect the U>T is a typo. I believe that standard sequencing methods would not detect U>T substitutions. It might also be helpful to clarify whether these are substitutions in the human strain using the bat strain as a reference or whether it is the other way around (see also other comments below). The following sentence “A similar trend was observed among the sequenced SARS-CoV-2 genomes,” should be removed. It was already stated that it applies to SARS-CoV-2 in general rather than a specific isolate.
Reply: The U>T was our mistake. It should read U>C. We apologize for this. The reference genome usage was clarified. The bat RaTG13 sequence was used as a reference in the bat to SARS-CoV-2 comparisons; human sequences were compared to the Wuhan Hu-1 isolate (MN908947). The sentence “A similar trend…“ was removed from the Abstract as recommended. We nevertheless keep to information about the outcome of the analysis of SARS CoV-2 variants saying “Accumulation of C>U mutations was also observed in SARS-CoV2 variants that arose in human population.”
- Line 33: “AT-rich” does not seem appropriate for an RNA virus.
Reply: corrected to AU-rich. Thank you.
- “preferring pyrimidine rich codons to purines” – is this specifically a property of the coding sequences or also the noncoding segments? This is important with respect to possible mechanisms leading to this preference.
Reply: The authors of this work meant representation of nucleotides in amino acid codons. In general, the non coding sequences are very short in viruses. For example, in CoV-2 they are less than 1 kb (<3% genome). Based on the referee’s question, we determined the purine and pyrimidine content in the whole genome and separately in both untranslated regions (Fig. S2). There does not seem a bias towards either py or pu nucleotides in the whole genome. The untranslated regions the ratios are disbalanced while the interpretation is difficult since the regions are short and lengths are not comparable with the coding regions. Interestingly the pu/py ratio is conserved between bat and human sequences which is in line with the prevalent transitions (which do not alter the pu/py ratio) over the transversions (changing the pu/py ratio).We mentioned the issue in the discussion.
- Line 75: A proper reference should be included. See the link “Citing Chimera” on the Chimera website.
Reply: The reference on the Chimera web site is included (line 96, ref. no. 24)
- Consolidate contradicting statements on lines 62 and 81. One states that SARS-CoV-2 was used as a reference, the other that RATG13 was used as a reference.
Reply: We thank the reviewer for the comment. The analysis of variation between bat and human included a bat sequences as a reference. The analysis of variation between SARS-CoV-2 isolates included the SARS-CoV-2 reference sequence Hu-1 from Wuhan (MN908947). We mentioned this in the Methods section that has now been elaborated and rewritten.
- Line 90: How was the chi square test used? What is the input for the test? I am asking because some of the key assumptions for the chi square test are that it is applied to categorical data (raw counts of observations, not percentages) and that all observations are independent. If counts of substitutions were pooled over multiple pairwise comparisons among different genomes then the assumption of independence is almost certainly violated because a single mutation can be observed in comparing different pairs of genomes and those do not represent independent events.
Reply: The comparative chi test was carried out in order to verify differences in mutation rates of individual nucleotides. While we think that it could be a suitable approach to analyse a single pairwise data as in the case of bat and human genome alignment we agree with the referee that it was wrong in the case of multiple pairwise comparisons over multiple pairwise comparisons it was a wrong approach. We decided to deal with the matter as follows. Instead of chi square tests we calculated the substitution rates as a ratio of observed frequency to expected frequency. The formulas used for the calculations are now described in the Methods section (page). The formula used for the expected frequency calculation considers unequal representation of nucleotides in the coronavirus genome (Figure 1a). The frequencies of probability of each nucleotide to change was normalized to the representation of each nucleotide in the coronavirus genome. Figure 1a shows that C is markedly underrepresented compared to other 3 nucleotides. The results are now presented in Figure 1b,c. It is evident that the under new calculations the C>U transitions are far more frequent that those of reverse (when going from bat to human sequence) and also compared to other substitution types.
- Line 91: “the overall frequency of transitions outnumbered the frequency of transversions by 5 fold” – how does this ratio of transitions and transversions compare to other types of organisms? Is it unusual?
Reply: The predominance of transitions over transversions is a general feature of prokaryotic and eukaryotic genomes. The extent may however differ between loci and organisms. The five fold difference we observe here is high compared to nuclear genomes. One of the explanations is that the transitions are mediated by C deaminations while transversions are not. In DNA, the C-deamination product - U is rapidly corrected by DNA repair processes. In contrast, RNA has no or limited correction system. Hence the ratio of transitions and transversions is likely to be higher in RNA genomes than in DNA genomes. Certainly, more studies are needed while the unlucky Cov-2 pandemic offers a convenient opportunity for the evolutionary studies that may help to address these questions.
- Figure 1: It is not clear how the direction of the substitutions was determined. In addition to conflicting statements about what the reference genome is, it is unclear if this is a comparison between RATG13 and a specific isolate of SARS-CoV-2 or some combination of comparisons between RATG13 and all SARS-CoV-2 isolates. I cannot imagine how it was done for panel b, which apparently summarizes many pairwise comparisons. It should probably be symmetrized by combining C>U and U>C substitutions etc. In any case, significant clarifications need to be added to the legend and to the description of the methods.
Reply: The sums of C>U+U>C and A>G+G>A counts are shown in Tables S3-4. We did not count C>U + U>C together in Figure 1 from the following reason: While in double stranded genomes this may be possible due to strand symmetry in organisms with single stranded genetic information such as coronaviruses this approach may not be entirely correct. In fact, when comparing bat and human sequences we observed 2 fold higher substitution rate of C>U than U>C which may be significant for thoughts about the virus evolution. We hope for providing a meaningful answer to your question.
- Figure 2: “The direction of mutations is from the human to bat sequence.” This is not very important but I think it would make more sense to use the bat isolate as a reference and the direction of mutations from bat to human. It is generally assumed that the virus jumped from bats to humans rather than the other way around, and one might argue that, in the absence of other evidence, the bat strain is likely to be more similar to their common ancestor than the human strains.
Reply: The direction of substitutions was adjusted to be from bat to human both in Figure 1b and Figure 3 (originally Figure 2). Please, note that the Figure showing a receptor domain alignment of bat and human sequences has been substantially revised.
- Table 1: Explain what is Ref./Position/Allele and Ref., Allele, and Shift. If Shift is the difference between Ref. and Allele than at least one number is incorrect (the line starting with 565). The legend states that the hydrophobicities come from the Kyte and Doolittle paper but in Methods they are claimed to come from the Chimera server.
Reply: The columns headings are now explained in the Table which is now Table S5 since we carried out a more detailed statistical analysis of amino acid characters in mutated codons and the box plots are shown in a new Figure 2. The Chimera server uses the experimental data of hydrophobicity values obtained by Kyte and Doolittle. Hence cite both Chimera and the Kyte and Doolitle paper.
- Line 123 and line 92: Why is the ratio of transitions to transversions 5 in one case and 2 in the other? I understand that these are different sets of data with the latter excluding the duplicate mutation counts but the difference can point to difficulties in estimating the mutation ratios accurately. I would expect that better methods of counting the mutations might produce more consistent results. I think these differences require more thought and reevaluating whether the methods are appropriate.
Reply: As mentioned in point 6 we recalculated the SNP frequencies in conceptually a new way considering the expected rates for each nucleotide. Though the outcomes using this and previous approach of substitution rates were comparable we think that the one presented in the current version of the ms is more accurate. The “observed to expected” transition ratios were about 5 when comparing human to bat sequences. Expressing the ratios in multiple comparisons is more difficult. We selected the approach in which all human sequences were pairwise compared to the same reference. Certainly, we are aware that the approach suffers from weaknesses since it is sensitive to the selection of reference. We nevertheless keep it in the paper mentioning the pitfalls. The reference genome was one of the first CoV-2 isolates while the other were sequences presumably derived from this putative ancestor (still not certain).
- Line 129: “We also determined the extent to which replacement would affect biochemical properties of replaced amino acids” – this is somewhat misleading because the only property that was investigated was the amino acid hydrophobicity.
Reply: The sentence was revised.
- Line 131: “The average change in hydrophobicity value [15] per C>U mutation was plus 3.68 while it was minus 2.85 for the reverse U>C mutation” – how do these values compare to other types of mutations? I am also puzzled why the values are different for C>U and U>C mutations. This, too, might warrant more discussion. The authors might also want to have a look at Tramontano and Macchiato 1986, Nucleic Acids Res.
Reply. The directional character of hydrophobicity change in C>U and U>C substitutions comes from the codons. In short, there are more C-rich codons for polar amino acids than are for hydrophobic acids. Conversely, there are U-rich codons for hydrophobic amino acids than are for polar amino acids. For example, in the cited blog article the authors calculated that “in of 48 C containing codons 17 C>U are synonymous, 27 nonsynonymous and 4 induce premature stop codons. The codon pool containing at least one cytosine codes for ten of the twenty amino acids, while the pool resulting from the 27 C>T transitions contains nine amino acids, where the mutant pool is almost twice as hydrophobic as the original”.
- The chi square statistics in Figure 3 may be problematic for the same reason I mentioned above. The Sars-CoV-2 genomes are nearly identical and can therefore hardly be considered independent for the purposes of the statistical test. I do not know how closely related the other viruses are but the same issue probably applies at least to some extent to the other groups as well. Applying the test to data that do not represent independent observations can dramatically inflate the perceived statistical significance. Along the same lines, the comparison in Figure 3a might be misleading if some of the groups are more diverse than others. For example, the SARS-CoV-2 genomes all appear consistently at the low side compared to the other groups but that could be a simple consequence of these viruses being much more closely related than those in the other groups.
Reply: We agree that the CpG differences may reflect phylogenetic differences between the groups while it was not our intention to place them into the phylogenetic context although it might be interesting. Phylogeny of coronaviruses may be complicated by the fact that they frequently change hosts. For example, the bat RaTG13 strain is more similar to human SARS-CoV-2 than to any other bat coronavirus. Perhaps, it was misleading to place SARS-Cov-2 outside of the human group. We therefore decided to include the SARS-Cov-2 reference genome with other human coronaviruses (left box). We arranged all human sequences and all bat sequences into separate groups and differences were statistically evaluated. Differences in CpG suppression are not significant (Pearson’s chi square test, P>0.05) between the groups.
- Line 158: I would question the phrase “strong transition-transversion bias typical for eukaryotic DNA genomes.” The bias is present in all domains of life and not typical of eukaryotes. Moreover, some viruses have been reported to have particularly strong transition-transversion bias, which might be worth mentioning in the discussion.
Reply: We accepted the suggestion and the transition/transversion bias was discussed and referencing the relevant literature.
- Line 159: The statement “Putative adaptive character of C>U mutations is demonstrated in the ACE2 receptor domain in which three out of five variable amino acids contained C>U signatures in their codons” is not supported by evidence. The authors mentioned that 3 of the 5 codons contain C>U or U>C mutations but left out that 4 of the 5 contain other mutations.
Reply: We wished to mention that we initially focused on six amino acids which constitute contact zone of the ACE2 receptor domain. Out of these, five variable amino acids differed between bat and human viruses. Out of the 7 nonsynonymous substitutions in these codons 3 had the C>U or U>C characters. While this can hardly be considered as an evidence for positive selection since the number of sites is too low. As recommended by the reviewer we therefore carried out a more comprehensive analysis of synonymous and nonsynonymous substitution in the s-glycoprotein gene which appears to be the most rapidly evolving part of the coronavirus genome (Fig. 3a and Table 1).
- Because, in the whole genome, the C>U/U>C accounts for 60% of all mutations, I think it is more reasonable to argue that the variability of the ACE2 receptor domain is mostly driven by other mutations than C>U/U>C. The authors also did not present any evidence that the receptor domain is under positive selection and thus directly involved in the adaptation to human host, although it is reasonable to expect that this is the case. It might be useful to compare Ka/Ks ratios in that region to that in the rest of the gene or in the other genes to support that claim. Performing Ka/Ks ratio analysis separately for C>U/U>C substitutions and other substitutions may also be relevant to how much the C.U/U>C mutations contribute to evolutionary adaptations of SARS-CoV-2 compared to other mutations, or whether they tend to be mostly neutral.
Reply: The reviewer is right in that the character of substitutions do not evidence a positive selection. To improve the analysis, as recommended, we carried out evolutionary tests using sequences encoding the SARS-CoV-2 s-glycoprotein sequences (please see a new Table 1). It appeared that the Ka/Ks ratios were below <1 implying there has been more synonymous changes than non-synonymous changes and that there has been evolutionary pressure to conserve the ancestral state - i.e. negative selection pressure. This holds true for the whole protein and the receptor binding domain subregion. However, the Ka/Ks values for the receptor binding domain were almost 10 fold higher (0.169) than in the rest of the protein (0.0268) suggesting that the nonsynonymous changes were more frequent in the receptor binding domain than elsewhere in the gene. In addition, three out seven nonsynonymous substitutions were C>U/U>C. These observations suggest the C>U substitutions did influenced evolution of this domain critical to species specificity of the virus. Certainly, most of C>U/U>C substitutions were neutral while considering their high number the chance of affecting amino acid sequence is relatively high. The issue of adaptive significance of any substitutions in not so easy to address particularly in systems with extremely high mutation rates. Coronaviruses similarly to other RNA viruses belong to this group. We summarized these thoughts in the Discussion subchapter “Do the C>U transitions have adaptive value?
- Line 161: “It is intriguing that a similar mutation trend was observed among 33 human isolates.” I do not see this as particularly surprising. I believe similar trends have been observed in HIV and other human viruses. The authors should include a review of relevant literature and compare this result with previously published data for other viruses.
Reply: Citations on transition biases in human viruses have been enlarged in numbers and included in the Discussion.
- Line 166: “the frequency of C>U substitutions was about 2 fold higher than that of reverse U>C substitutions” – I am very skeptical about this statement considering that the authors did not explain how they determined the direction of the mutations and that they apparently count a single mutation multipole times. Notably, the C>U and U>C mutations are about equally frequent in Figure 1a, which again is left out from this argument.
Reply: We understand the reviewer’s concern over the issue. We realize that the Figure 1a was confusing and the way we presented the data made the conclusion made. We apologize for that. In order to better support the conclusions we expressed the substitution rate as a ratio between observed to expected counts following equations in the Methods section. In a bat to human virus comparisons the observed/expected substitution rates were 5 and 2.7 for C>U and U>C, respectively. The substitution rate of C>U substitutions seems to be, indeed, almost 2 fold higher than that of reverse U>C substitutions. We believe that it may be evolutionary signficant.
- The argument on lines 170-176 may be problematic. First of all, no evidence is presented to support the statement “the frequency of C>U (U>C) transitions between the SARS-CoV-2 genomes is consistent with spontaneous C deamination events.” Moreover, the estimate of one C deamination per 200 years is not specific to SARS-CoV-2 and the mutation rates could be very different in viruses and cellular organisms, and also very different among different viruses.
Reply: The reviewer is right that there might other explanations of high mutation rate in RNA viruses. We included the possibility that the C>U/U>C transition biases are caused by polymerase errors during virus replication (J Transl Med 2020, 18, (1), 179) and developed the discussion in this way. Nevertheless, the C deamination events seem to explain the observation the best. Based on known C deamination rate we also calculated the probability of C>U mutations which appeared to be slightly higher than that of estimated mutations rates of coronaviruses. This observation has been discussed in the Discussion section. The sentence ““…the frequency of C>U (U>C) transitions between the SARS-CoV-2 genomes is consistent with spontaneous C deamination events.” has been replaced by a more neutral one.
- Line 180: “Some transitions markedly altered biochemical properties of amino acids evidencing for a positive selection at specific sites” – that some transitions changed hydrophobicity of the encoded amino acid can in no way be considered evidence of positive selection. Additional evidence is needed to make that claim. A detailed Ka/Ks ratio analysis with appropriate assessments of statistical significance might be helpful in that regard. See Choi and Lahn 2003, Genome Res., for an example how it can be used.
Reply: As mentioned in point 17, we carried out the analysis of synonymous and nonsynonymous substitutions in the whole spike glycoprotein genes and its subregion encoding an ACE2 receptor binding domain. Please, see a new Table 1 for the details
- Line 183: “consistent with shifts towards codons for more hydrophobic amino” – this is another problematic statement. There is no indication that the authors made any attempt to determine the ancestral sequence and the actual direction of the mutation or any particular C-U mismatch in the alignment. Consequently, it is impossible to tell whether the mutation was C>U or U>C and whether the amino acid change was from less hydrophobic to more hydrophobic or the other way around.
Reply: The reviewer is right that this part was sloppily described in the original version. We apologize for this. To improve the presentation of results we carried out a statistical analysis of C>U, U>C and other mutations using the data sets comprising all SARS-CoV-2 variants (Please see a new Figure 2 and Table S5). It appeared that the differences between the groups were highly significant: The C>U substitutions increased the hydrophibicity values while those of U>C were decreasing it when compared to other mutations. Further the authors of the web blog “Cytosine deamination and evolution” calculated that there are 48 possible substitutions in the C-containing amino acids codons. Of these, 17 are synonymous, 27 nonsynonymous and 4 introduce stop codons. The pool resulting from the 27 C>U (in the original work U was replaced with T) substitutions contains nine amino acids, where the mutant pool in almost twice as hydrophobic as the original. The opposite trend is indicated with the reverse U>C mutations. These observations suggest that the random C deamination events (either spontaneous or enzymatically catalysed) during the virus life cycle may indeed induce changes in amino acid composition. These mutations could be fixed or disappear depending on selection. It is our assumption that most of them actually disappear while a minority may change virus properties (e.g. adaptation to a new host) and become fixed.
- Line 186: “the virus protein products would be modified by ongoing and relatively fast C deamination processes affecting its genome” – as stated above, it is unclear if the C>U mutations contribute to nonsynonymous changes in proteins more than other mutations. At the minimum, the authors should include other types of mutations in Table 1 and compare the synonymous-to-non-synonymous ratios for different types of mutations. There is also still the issue related to uncertainty as to the actual direction of the mutation, which could only be determined if the ancestral allele is known.
Reply: The reviewer raised an important question in this point. Unfortunately the ancestral type of SARS-CoV-2 is unknown although it is generally believed that the reference Wu-Hu-1 sequence is most close to the ancestral type since it originates from the first Cov-2 isolate in Wuhan. To address the issue of C>U mutations underlying synonymous and nonsynonymous mutations we carried out the analysis of spike glycoprotein including its receptor binding domain subregion. The results presented in Figure 1d, e. and Table 1 suggest that though the number of nonsynonymous C>U/U>C mutation is generally low (Fig. 1d) their proportion is relatively high (Fig. 1e) in positions encoding critical amino acids needed for a virus contact with the cell.
- One important thought related to CpG suppression is that, in RNA, the likelihood of C>U substitution should be independent of whether the C is preceded by G or not. The reason why C is much more likely to change to T in some organisms’ DNA if it is preceded by G is that the C is first methylated and a subsequent deamination changes the methyl-C to T. An unmethylated C changes to U upon deamination, and the U is generally enzymatically converted back to C when it is in DNA. Consequently, processes operating on RNA might be responsible for the general C suppression whereas process operating on DNA might be responsible for CpG suppression. In that regard, it might be interesting to include more discussion on how these results relate to the physiology of coronaviruses (or other RNA viruses), particularly with respect to whether they persist in the host in the DNA form.
Reply: We thank the reviewer for her/his stimulating remark. We believe that the CpG depletion in both DNA and RNA genomes has a similar molecular basis which are the C>T and C>U transitions in DNA and RNA, respectively. The magnitude of CpG suppression differs however largely between the genomes and genomic regions. In highly methylated and repeated rich nuclear genomes the CpG suppression is explained by cytosine methylation which is known to be concentrated in non coding regions. Consequently, pseudogenes are often A+T rich compared to functional genes. The CpG suppression is more difficult to explain in RNA viruses which probably do not methylate their genome (although this is just an assumption not properly studied) and do not harbour a significant fraction of non coding RNA. In SARS-CoV-2 the non coding fraction constitutes less than one percentage of the genome. Yet, the frequency of CpG and GpG dinucleotides is 0.015 and 0.037 indicating CpG suppression. However, both CpG and GpC are clearly the least represented motifs among the 16 dinucleotide pools (Figure S2). We favour the hypothesis that coronavirus for which no DNA form is known are exposed primarily to C deamination events which may some specificity to particular dinucleotides. Perhaps, cellular cytidine deaminase(s) might have higher activity to Cs flanked by Gs at their 3’. Alternatively, Cs in CpG dinucleotides could be chemically less stable than Cs in other contexts. We mentioned these possibilities in the Discussion.
- More attention might also be given to the observation of elevated A>G and G>A mutations. It suggests a possible mechanism involving C deaminations in the complementary strand, which again can take place only in a double-stranded nucleic acid.
Reply: The reviewer is right in this point. In nuclear DNA all transitions are usually balanced. In coronaviruses (both bat and human) it was puzzling that the A>G and G>A substitutions were much less frequent than those of C>U and U>C. In fact, the ratio between observed and expected was 1.2 and 1.7 for A>G and G>A, respectively which is at the edge of significance. How, to explain this imbalance? We think that the heart of the transition mutations imbalance lies in the virus replication cycle. Coronaviruses spend most of their life in a single stranded positive RNA form and complementary negative strand is produced only during virus replication. It is estimated that these negative-strand intermediates are only about 1 % as abundant as their counterparts (J Virol 1991, 65, (1), 320-5). As a consequence, a negative strand of the virus genome is less likely to accumulate C>U transition than the positive strand. Hence, the G>A substitutions are relatively rare compared to those of C>U although both transitions are more common than any of the transversions.
- The final conclusion (lines 216-218) could be more accurately rephrased in the sense that C>U mutations driven by spontaneous cytosine deamination may contribute to rapid evolution of the virus. I believe this is supported by the data in the paper but some of the more specific statements made earlier are not.
Reply: This has been done in the last paragraph 4.3. We wrote: Several hypotheses have been proposed to explain the origin of SARS-CoV-2 virus [14]: (i) natural selection in the animal host before zoonotic transfer; (ii) natural selection in humans following zoonotic transfer; (iii) the virus is a product of artificial manipulation. Our data make the third possibility less likely, since the viral genome seems to follow a similar evolutionary trajectory as that of the closely related bat virus RaTG13. We favor the second possibility postulating that cytosine deamination processes are driving the SARS-CoV-2 genome evolution, particularly over short evolutionary times. Resulting C>U biases in mutation spectra could be a stigma of virus adaptation in its host(s).
- Supplementary tables need explanatory legends.
Reply: Legends were added to the Tables placed as footnotes or in the right margins in complexed data sets.
- I would urge authors to provide more context for each mutation listed in Table S3: whether it is in a gene or intergenic, synonymous or non-synonymous, the amino acid change, …
Reply: The details of each mutation in human variants of SARS-CoV-2 is now given in Table S5. The Table contains information about the position of SNV in the genome, mature peptide, untranslated region, the amino acid change and hydrophobicity values of altered amino acids.
Typos and other recommended minor corrections:
- Abstract: pandemy -> pandemic; “interest about its evolutionary origin” -> “interest in its evolutionary origin”; italicize “Rhinolophus affinis”; “Contrast to most other coronaviruses both SARS-CoV-2 and RaTG13 exhibited CpG depletion in their genomes” -> “In contrast to most other coronaviruses, both SARS-CoV-2 and RaTG13 exhibited CpG depletion in their genomes”; “The data support that the hypothesis” -> “The data support the hypothesis”
Reply: Accepted.
- Line 39: “completely sequence” -> “completely sequenced”
Reply: Accepted
- Line 60: “intragenomic variation between 60 coronaviruses”: if anything, it should probably be “intergenomic” but I would recommend “sequence variation”
Reply: Accepted.
- Line 65: “were mapped the reference genome” -> “were mapped [on|to|onto] the reference genome”
Reply: Accepted.
- Line 70: “the number of observed CpG/number of expected CpG dinucleotides” -> “the ratio of observed and expected counts of CpG dinucleotides”
Reply: Accepted
- Line 76: “On line” -> “Online”
- Line 84: “resulting to” -> “[leading to|resulting in|reflecting]”
- Line 92: “by 5 fold” -> “5 fold”
- Line 140: “between the genomes” -> “among the genomes”
- Line 156: “despite these represent” -> “despite these representing”
Reply: Lines 6-10. Accepted
Reviewer 2 Report
Matyášek and KovaÅ™ík report a strong bias towards C>U, U>C, A>G and G>A mutations iin SARS-CoV-2 and the related bat CoV RATG13 genomes. More than 80% of mutations can be attributed to these four substitutions despite representing only one third of all possible substitutions. This is an interesting finding that will be of interest to evolutionary virologists.
Author Response
Dear Sir/Madame,
We wish to thank you for your kind reading our ms. We are glad you find it interesting and worth publishing,
In order to improve the English language and style the revised version has been corrected by the MDPI editing service.
Sincerely,
Ales Kovarik
Round 2
Reviewer 1 Report
The authors addressed most of my comments adequately. However, the key problem related to determining the direction of mutations has not been resolved. I think this is a critical issue for correct presentation and interpretation of the data and I believe the authors may have misunderstood my comment. This is a more elementary explanation of what I am concerned about:
Let’s assume this alignment between sequences from Genome1 and Genome2:
Genome1: AAGCAAG
Genome2: AAGUAAG
The mismatch at position 4 may have arisen via different scenarios with the first two being most parsimonious:
- If the ancestral sequence had C at position 4, a C>U substitution occurred in Genome2.
- If the ancestral sequence had U at position 4, a U>C substitution occurred in Genome1.
- If the ancestral sequence had another nucleotide X (A or G) at position 4 then an X>C mutation occurred in Genome 1 and X>U mutation occurred in Genome2. This scenario can be rejected as unlikely based on parsimony principle (as well as other scenarios involving two or more substitutions) and I will not consider it further but I am including it for completeness.
The key is that in the absence of knowledge of the ancestral sequence, it is impossible to tell if this alignment represents a C>U substitution that took place in Genomr2 or a U>C substitution that took place in Genome1. There is no indication that the authors attempted to determine the ancestral sequence and they admit themselves that it would be difficult to do (although it can be done with some level of uncertainty using multiple sequence alignments). I suspect that their distinction between C>U and U>C substitution depends solely on which genome was designated Genome1 and which was designated Genome2, or in other words, on the order in which the genomes were compared. That is an arbitrary decision that does not reflect a true direction of the substitution. As I stated in my original review, for comparisons between bat and human isolates, it might be justifiable to assume that, in most instances, the bat allele is the ancestral allele based on the generally accepted view that the virus transferred from bat to humans, but one should be aware that this may not be true for all mutations and this assumption and its justification should be discussed in the paper. However, when comparing two human isolates, there is far less support for assuming one to be more similar to the ancestral sequence than the other and as a result one cannot determine whether the mutation is C>U or U>C. In such instances, I believe the counts of C>U and U>C mutations cannot be separated and the text and data presented in the paper need to be appropriately revised. If the authors disagree they should explain how they determine or predict the true direction of the mutations without relying on arbitrary criteria or unjustified assumptions.
Author Response
We thank the reviewer for a very didactical explanation on what may be happening during molecular evolution of sequences. We are aware of the fact the directionality of substitutions is dependent on the availability of an ancestral sequence from which all sequences are derived and which should ideally be used a reference. However, identification of the ancestral sequence in rapidly mutating viruses is not an easy task since the ancestral sequence is short-lived. Considering coronavirus genome mutation rate of ten to the minus fourth per replication cycle it is about 4 orders of magnitude higher than the mutation rate of mammals. In other words, virus (RNA plus) strains which infected organisms few years or even months ago do not have identical genomes with those of present strains. Thus identification of an ancestral genome of SARS-CoV-2 is a difficult task and results are hampered by considerable uncertainty.
Based on the reviewer's suggestion we nevertheless attempt to determine the phylogenetic relationships between human SARS-CoV-2 isolates using maximum-likelihood algorithm. As revealed in a new Figure 2 three branches were revealed two of which (S and L clades) corresponded to published genotypes. We mapped mutation characters to the phylogeny. It appeared that the S branch considered to be most close to the animal progenitor (National Science Review 2020, 7, (6), 1012-1023, https://academic.oup.com/nsr/article/7/6/1012/5775463) contained relatively low proportion of C>U variants compared to the S’ and L groups which may support the hypothesis that the C>U transitions are indeed accumulating during the pandemic.
Because of apparent uncertainty in determining the SARS-CoV-2 ancestral genotype we expressed the substitution rates as the forward and reverse SNPs, i.e C>U + U>C, as recommended by the reviewer (please see revised Figures 1b,c). We just mentioned in text the fact that the C>U transitions were more frequent than those of C>U while toning downing strong statements.
Finally, we wish to thank the reviewer for her/his apparent interest in reading of our ms and helpful comments.